# Neuropeptide signalling shapes feeding and reproductive behaviours in male *Caenorhabditis elegans*

Matthew J Gadenne[1],* , Iris Hardege[2],* , Eviatar Yemini[3] , Djordji Suleski[1] , Paris Jaggers[2] , Isabel Beets[4] , William R Schafer[2,4] , Yee Lian Chew[5]

**Sexual dimorphism occurs where different sexes of the same species display differences in characteristics not limited to reproduction. For the nematode *Caenorhabditis elegans*, in which the complete neuroanatomy has been solved for both hermaphrodites and males, sexually dimorphic features have been observed both in terms of the number of neurons and in synaptic connectivity. In addition, male behaviours, such as food-leaving to prioritise searching for mates, have been attributed to neuropeptides released from sex-shared or sex-specific neurons. In this study, we show that the *lury-1* neuropeptide gene shows a sexually dimorphic expression pattern; being expressed in pharyngeal neurons in both sexes but displaying additional expression in tail neurons only in the male. We also show that *lury-1* mutant animals show sex differences in feeding behaviours, with pharyngeal pumping elevated in hermaphrodites but reduced in males. LURY-1 also modulates male mating efficiency, influencing motor events during contact with a hermaphrodite. Our findings indicate sex-specific roles of this peptide in feeding and reproduction in *C. elegans*, providing further insight into neuromodulatory control of sexually dimorphic behaviours.**

## Introduction

Sexual dimorphism is where two sexes of the same species show differences in behaviour or anatomical features not limited to reproduction, such as body size or muscle mass. These sex differences are important to promote organismal reproductive fitness and survival (Fairbairn, 1997). In the mammalian nervous system, sexual dimorphisms arise because of a combination of sex differences in the genome, hormonal influence, and differences in signalling within and between neural circuits (McCarthy & Arnold, 2011). It is not yet well-understood precisely how sexual dimorphism arises in the brain, or how these contribute to differences in

behaviour or in the incidence of some neurological conditions (Clayton, 2016). Increasing evidence suggests that neuromodulator signalling contributes to these sex differences (see, e.g., Liu et al [2007], Asahina et al [2014], and Tabatadze et al [2015]); however, the size and complexity of the mammalian nervous system means that there is a relatively limited mechanistic understanding of neuromodulator networks, and how they interact with neural circuits, in these organisms. Investigating the specific effects of neuromodulators on sex differences in the brain and on behaviour in a more compact and tractable system may help clarify the general principles through which these internal signalling mechanisms regulate sexual dimorphism.

The two sexes of *Caenorhabditis elegans* are anatomically and behaviourally distinct. Hermaphrodite gonads produce both sperm and eggs, allowing them to self-fertilise and reproduce independently of male worms. Hermaphrodites do not mate with other hermaphrodites. In contrast, as males only produce sperm, male reproduction is dependent on mating with a hermaphrodite. Male worms also have more neurons, including 91 male-specific neurons, many of which are involved in coordinating reproductive behaviours (Barr et al, 2018). In *C. elegans*, neuroanatomical differences between the sexes arise from a combination of sex-specific programmed cell death, differentiation, and neurogenesis (reviewed in Portman [2017] and Barr et al [2018]).

In addition to sexually dimorphic features in neural circuits, including differences in synaptic connectivity (Cook et al, 2019; Bayer et al, 2020), neuromodulators can act differently in the two *C. elegans* sexes to drive sex-specific behaviours (Barrios et al, 2012; Reilly et al, 2021). One such class of neuromodulators, called neuropeptides, have been shown in multiple experimental models to be required for broad modulatory actions (Bargmann, 2012), because of their ability to trigger responses "extrasynaptically" between neurons not physically connected by synapses (Bentley et al, 2016). *C. elegans* provides an excellent system to study how neuropeptides control sex-specific behaviours: it is genetically tractable, amenable to a diverse technological toolbox for

[1]Molecular Horizons, University of Wollongong and Illawarra Health and Medical Research Institute, Wollongong, Australia   [2]Neurobiology Division, MRC Laboratory of Molecular Biology, Cambridge, UK   [3]Department of Neurobiology, UMass Chan Medical School, Worcester, MA, USA   [4]Department of Biology, KU Leuven, Leuven, Belgium   [5]Flinders Health and Medical Research Institute and College of Medicine and Public Health, Flinders University, Adelaide, Australia

Correspondence: yeelian.chew@flinders.edu.au
*Matthew J Gadenne and Iris Hardege contributed equally to this work.

interrogation of the nervous system, and has a largely complete neuronal connectome in both sexes (White et al, 1986; Walker et al, 2017; Cook et al, 2019; Witvliet et al, 2021). Understanding how neuropeptides modulate sexually dimorphic behaviours in *C. elegans* may provide a platform to investigate how similar modulatory systems function in bigger brains.

In animals that reproduce sexually, including *C. elegans*, two key appetitive behaviours are feeding/food-seeking and mate-seeking. Male *C. elegans* experience both a reproductive pressure of having to find mates to pass on genetic traits, as well as a competing "feeding" selective pressure that drives them to seek abundant food (Ryan et al, 2014; Wexler et al, 2020). A characteristic behaviour of male worms is that in the presence of food but absence of mates, males leave food to prioritise mate searching (Lipton et al, 2004; Barrios, 2014). This behaviour is regulated by neuropeptide signalling from both sex-shared and sex-specific neural circuits (Barrios et al, 2012; Garrison et al, 2012).

Here, we investigated the role of a specific neuropeptide, encoded by the gene *lury-1* (luqin RYamide), in modulating mating and food-seeking behaviours in *C. elegans* males. A previous study of LURY-1 peptides in *C. elegans* focused on phenotypes in hermaphrodite animals, showing that these peptides inhibit feeding and stimulate egg-laying in a food-dependent manner through interactions with the neuropeptide receptor NPR-22 (Ohno et al, 2017). In hermaphrodites, these LURY-1–dependent behaviours are triggered by peptide release from pharyngeal M1 and M2 neurons. In this study, we show that *lury-1* has a different expression pattern and impact on feeding behaviour in the two sexes. In addition, LURY-1 signalling also modulates mating behaviour in male worms. Our results identify for the first time a sexually dimorphic role of LURY-1 neuropeptides in *C. elegans*.

## Results

When investigating the expression pattern of *lury-1* using a polycistronic fluorescence reporter line (P*lury-1(3.4 kb)::lury-1 gDNA + UTR::SL2-mKate2*), we discovered additional cells expressing this reporter in male *C. elegans* compared with hermaphrodites (Figs 1A and S1). As previously reported, expression in hermaphrodites is limited to pharyngeal neurons, identified as M1 and M2 neurons (Ohno et al, 2017) (Fig 1A). In males, expression of *lury-1* is observed in the pharyngeal neurons, as well as in cells of the male tail (Figs 1A and S1). We observed additional (dimmer) expression of *lury-1* in the M5 pharyngeal neuron (Fig S1), identifying this neuron via its dorsal-left-posterior quadrant in the posterior pharyngeal bulb and ventral projection. No expression in the hermaphrodite tail was observed in our study (Fig 1A) or in previous reports (Ohno et al, 2017). We identified the *lury-1*–expressing neurons in the male tail using the recently published tool NeuroPAL (Tekieli et al, 2021; Yemini et al, 2021). As NeuroPAL can only be used together with green fluorescence reporter lines, we used the P*lury-1*::Venus line from (Ohno et al, 2017) to perform these analyses. We identified DX1, DX2, and DX3 as the *lury-1*–expressing neurons in the male tail (Figs 1 and S1). DX1 and DX2 are in the dorsorectal ganglion whereas DX3 is in the preanal ganglion. In a recent whole-brain imaging study of

male worms, DX1, DX2 and DX3 were shown to become active specifically during the "turning" phase of mating behaviour (Susoy et al, 2021), suggesting a role in coordinating movement during the male mating process. DX neurons have also been implicated in regulating sperm transfer and release (Liu, 1995). We further explored the role of these *lury-1*–expressing neurons by examining their synaptic connectivity (Jarrell et al, 2012; Cook et al, 2019). DX1–DX3 neurons all have gap junctions to the dorsal body wall muscle and are connected via synapses to multiple neurons in the male tail. Interestingly, DX1, DX2, and DX3 neurons are synaptically connected to HOA and HOB (all presynaptic to HOA, all postsynaptic to HOB), sensory neurons that form part of the hook sensillum in the male tail (Cook et al, 2019). Previous research indicates that HOA/HOB is required to sense the hermaphrodite vulva during mating (Liu & Sternberg, 1995). DX1, DX2, and DX3 neurons are also connected to ray neurons, including R3BL/R, R9AL, and R9AB; these male-specific sensory neurons contribute to "apposition behaviour" where the male tail makes and maintains contact with the hermaphrodite during scanning and turning (Barrios et al, 2008; Koo et al, 2011; Susoy et al, 2021). Lastly, DX1-DX3 neurons are connected to neurons that innervate the post-cloacal sensillum, being presynaptic to PCA (all) and PCC (DX3) neurons, and receiving inputs from PCB (DX1/DX3). PCA/PCB/PCC are sensory/motor-neurons that regulate spicule insertion and sperm release (Garcia et al, 2001; LeBoeuf et al, 2014). Taken together, these expression and connectivity data indicate that *lury-1*–expressing neurons in the male tail are likely to be involved in coordinating male-specific mating behaviours.

Neuropeptides in *C. elegans* are thought to act predominantly through G protein-coupled receptors (GPCRs) (Frooninckx et al, 2012). As first demonstrated in Ohno et al (2017), we independently found that LURY-1 neuropeptides can activate the GPCR NPR-22 with high potency (dose response curve and EC50 values indicated in Fig 1C) using an in vitro screening protocol (Beets et al, 2012). The expression pattern of NPR-22 in hermaphrodites was previously reported in Turek et al (2016) and Palamiuc et al (2017) and includes expression in the pharynx and head neurons (Fig 1D). Using a GFP-tagged fosmid reporter line for NPR-22, we found that expression in male *C. elegans* is observed in the pharynx, and additionally in cells of the male tail including hypodermal support cells and potentially the hook structure (Fig 1D). These data suggest that LURY-1 could signal to NPR-22 in the pharynx to modulate feeding, and to NPR-22 in male tail structures to modulate copulation.

Given the sex-shared expression of *lury-1* in excitatory pharyngeal neurons known to regulate pharyngeal muscle movements (Pilon & Morck, 2005), we monitored pharyngeal pumping in both hermaphrodites and males (Fig 2A). Ohno et al (2017) showed that LURY-1 impacted feeding in hermaphrodites, with *lury-1* over-expression leading to decreased pharyngeal pumping; suggesting that LURY-1 signalling suppresses feeding behaviours (Ohno et al, 2017). In hermaphrodites, we found that mutant animals carrying the *lury-1(gk961835)* allele, a putative null allele that deletes most of the first exon, showed increased pharyngeal pumping (Fig 2B). This implies that LURY-1 acts to suppress pharyngeal pumping, consistent with the findings of Ohno et al (2017). We did not observe

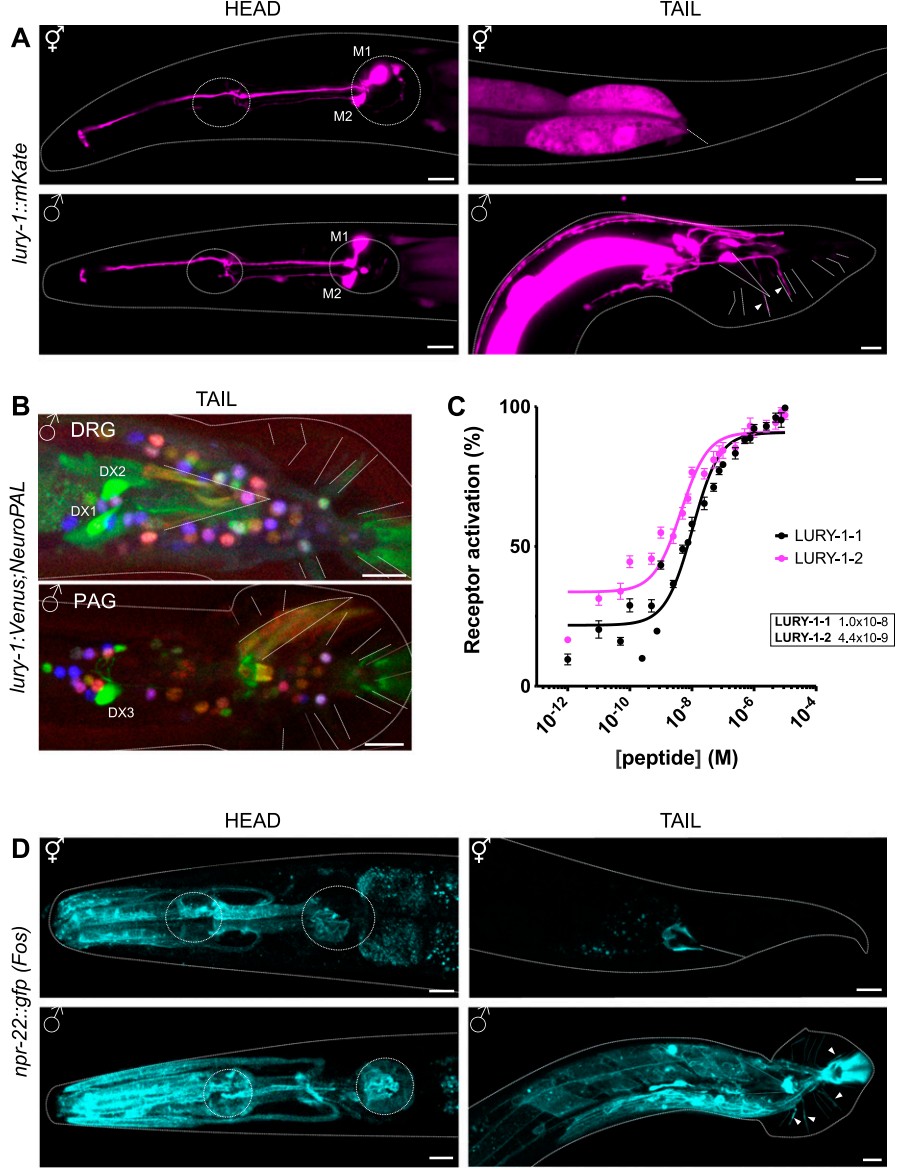

**Figure 1. Neuropeptide *lury-1* and its receptor *npr-22* are expressed in different cells in the two sexes of *C. elegans*.**

**(A)** A reporter line expressing the transgene P*lury-1(3.4 kb)::lury-1 genomic DNA::SL2-mKate2* shows expression in pharyngeal neurons M1 and M2 in both hermaphrodites (top) and males (bottom). In males, there is additional expression in cells in the tail (white arrows). Dashed outlines indicate worm body and position of pharyngeal bulbs in the head. Scale bar = 10 μm. **(B)** Identification of *lury-1*–expressing male tail neurons using NeuroPAL (Tekieli et al, 2021; Yemini et al, 2021). The *lury-1* reporter line used here is JN2443 (P*lury-1::Venus, unc-122p::mCherry*). The top panel shows DX1 and DX2 in the dorsorectal ganglion (DRG), the bottom panel shows DX3 in the preanal ganglion (PAG). DX1–DX3 are annotated in the image. Nuclear fluorescence in other neurons is in pseudocolour as in Yemini et al (2021). Scale bar = 10 μm. **(C)** Dose–response curve showing that peptides encoded by the LURY-1 precursor (LURY-1-1 and LURY-1-2) activate the receptor NPR-22 in vitro. The corresponding EC50 values (M) for each peptide are as indicated. Line represents non-linear regression fit of a variable slope line using three parameters. n = 6–8 trials. **(D)** A reporter line expressing a fosmid containing C-terminal GFP-tagged *npr-22* shows expression in pharynx muscle and cells of the tail in both hermaphrodites (top) and males (bottom). Expression in male copulatory apparatus is also observed. Dashed outlines indicate worm body and position of pharyngeal bulbs in the head. Scale bar = 10 μm.

a pharyngeal pumping effect in *npr-22(ok1598)* mutant hermaphrodites (Fig 2B), a null mutation with a 2.5 kb deletion in the *npr-22* genomic locus (deleting four of six exons). In contrast, male *lury-1* and *npr-22* mutant animals show decreased pharyngeal pumping compared with controls (Fig 2C). Our findings suggest that LURY-1 signalling in males increases pharyngeal pumping, which is opposite to that observed in hermaphrodites. This is curious as pharyngeal neuron expression of *lury-1* is shared between the two sexes, and yet the pharyngeal pumping phenotype of *lury-1* mutants is sexually dimorphic.

Like the hermaphrodite experiments shown in Fig 2B, we also tested pharyngeal pumping rate in male animals overexpressing *lury-1*. In these experiments, we found that *lury-1* overexpression showed reduced pharyngeal pumping in male worms, similar to what was observed in hermaphrodites (Fig 2D). These data suggest that, unlike the null mutants, increased gene copy

number of *lury-1* has the same effect on pharyngeal pumping in both sexes.

LURY-1 signalling in hermaphrodites modulates egg-laying behaviours (Ohno et al, 2017), indicating a role in reproduction. As we found LURY-1 expression in the specifically in the tail of male *C. elegans* (Fig 1B), in neurons associated with regulation of mating behaviour (Susoy et al, 2021), we asked if male reproduction was regulated by LURY-1. To assess male mating efficiency, we picked *fog-2* mutant hermaphrodites, which produce eggs but no sperm—and therefore cannot self-fertilise—onto individual plates with a single male worm (or no male), removing the male after a few hours (see the Materials and Methods section). If no male is present (annotated "*no male control*"), or if the male does not mate with the *fog-2(–)* hermaphrodite, there will be no progeny produced (Fig 3Ai and B). If the male successfully mates with the *fog-2(–)* hermaphrodite, there will be progeny produced through cross-fertilisation (Fig 3Aii).

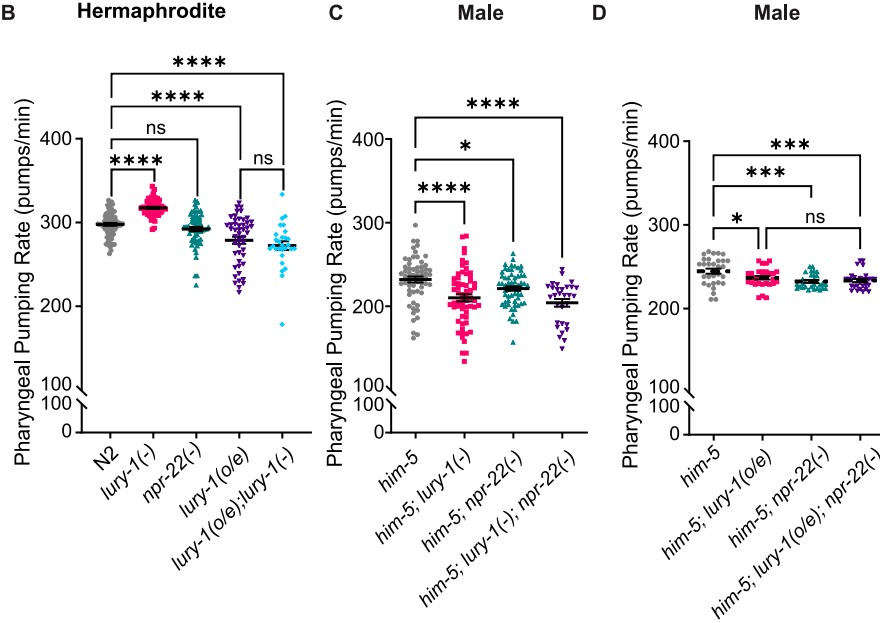

Figure 2. **LURY-1 peptides modulate pharyngeal pumping differentially in hermaphrodites and males.**
**(A)** Pharyngeal pumping assays were performed on well-fed animals, with movements of the pharynx grinder counted per minute (pumps/min). **(B, C, D)** Pharyngeal pumping rate of (B, C) *lury-1* and *npr-22* mutant strains and (B, D) *lury-1* overexpression (2 kb promoter) transgenic lines containing *lury-1* or *npr-22* mutant alleles. **(C, D)** Experiments with male *C. elegans* (C, D) were performed in the *him-5*–mutant background. n > 10 per replicate for >3 biological replicates. Scatter plots show all data points, with error bars indicating mean ± SEM. *P*-values indicated by ns = not significant. *P < 0.05, ***P < 0.001, ****P < 0.0001 (one-way ANOVA with Fisher's post-test).

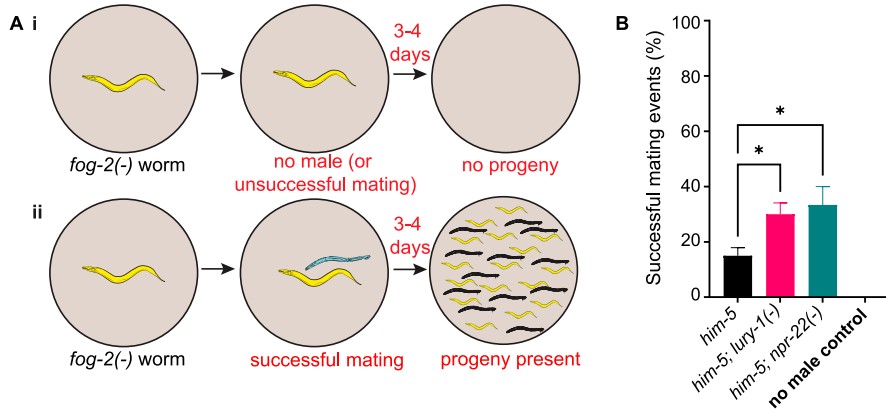

Figure 3. **LURY-1 peptides modulate male mating efficiency.**
**(A)** Male mating efficiency assay: (i) *fog-2(–)* produce eggs but no sperm, so will not produce progeny if mating with a male does not occur, or if no male is present. (ii) If mating successfully occurs, then progeny will be observed after several days. **(B)** Mating efficiency for *lury-1* and *npr-22* mutant males left to mate with *fog-2(–)* hermaphrodites for 3 h. Experiments were performed in the *him-5*–mutant background. n > 10 males per genotype, per replicate for >3 biological replicates. Error bars indicate mean ± SEM. *P*-values indicated by ns = not significant. *P < 0.05 (one way ANOVA, Fisher's post-test).

We found that *lury-1* and *npr-22* mutant males (both crossed into the *him-5* background) showed more successful mating events than control males (Fig 3B). This suggests that LURY-1 signalling suppresses male mating, potentially via NPR-22. We also tested the impact of *lury-1* overexpression on male mating; however, no statistically significant effect of overexpression was observed compared with control males (Fig S2). One possible explanation for this is that transcription of the *lury-1* gene may not be the rate-limiting step in the modulation of male mating, and that post-transcriptional processing or peptide release may be the

stage at which regulation of LURY-1 signals are critical in this context.

Because *lury-1*–mutant males showed a higher mating efficiency compared with controls (Fig 3B), we next asked whether LURY-1 affected individual male copulation behaviours. The *C. elegans* mating process is a complex and stereotyped series of behaviours, where the male worm makes contact with the hermaphrodite and scans for the vulva. If it fails to find the vulva, it may turn over the hermaphrodite head or tail to scan the other side of the hermaphrodite body (Liu & Sternberg, 1995) (Fig 4Ai). Previous

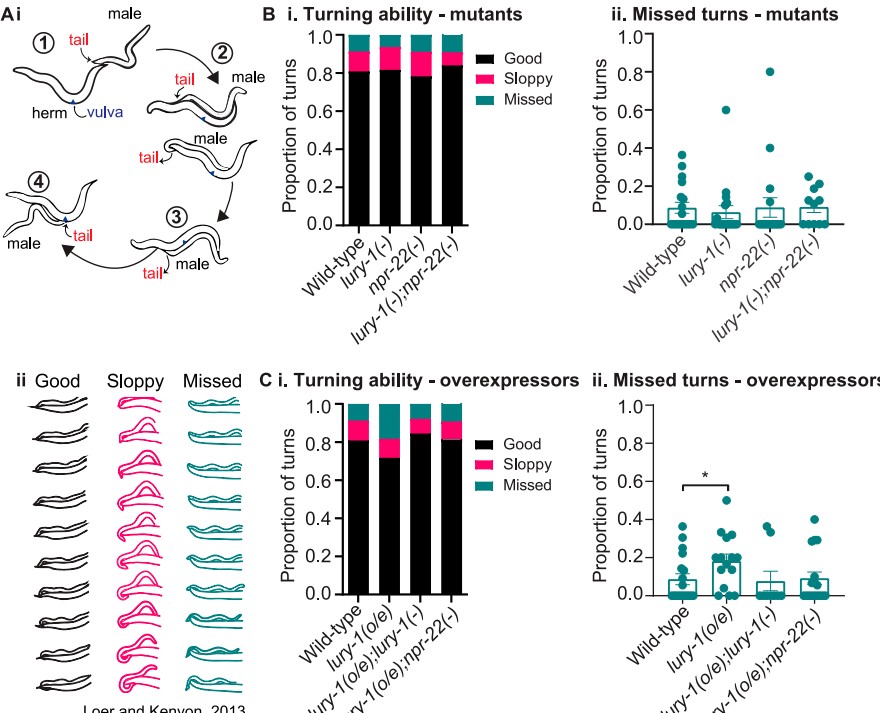

**Figure 4. LURY-1 overexpression leads to less efficient turning during male mating.**
**(A) (i)** Male mating behaviour is a complex multi-step process that begins with (1) contacting the hermaphrodite with the male tail and backward movement in search of the vulva; (2) turning and scanning: the male presses its tail against the body of the hermaphrodite and moves backwards until it reaches the vulva. If the male does not detect the vulva, it will make a tight turn and search along the other side; (3) once the vulva is located, the male will prod the slit with his spicules; (4) the male locks to the hermaphrodite and ejaculates his sperm inside. **(ii)** Illustrations of good, sloppy, and missed turns, adapted from Loer and Kenyon (1993). **(B) (i)** Proportion of good, sloppy, and missed turns and **(ii)** missed turns as a proportion of all turns as observed in video recordings as observed in video recordings for *lury-1*– and *npr-22*–mutant males left to mate with an *unc-13* hermaphrodite. Data from B(ii) are extracted from the graph in B(i). From left to right, n = 18, 18, 17, and 11. Error bars = mean ± SEM. All comparisons are not statistically significant. **(C) (i)** Proportion of good, sloppy, and missed turns and **(ii)** missed turns as a proportion of all turns for *lury-1* overexpression transgenic lines crossed with *lury-1*– or *npr-22*–mutant males left to mate with an *unc-13* hermaphrodite. Data from C(ii) are extracted from the graph in C(i). From left to right, n = 18, 15, 9, and 17. Error bars = mean ± SEM. *P*-values indicated by *<*P* = 0.05 (one-way ANOVA with Fisher's post-test).

reports have carefully characterised the male turning behaviour and categorised it into three (3) types: good, sloppy, and missed (Loer & Kenyon, 1993). Good turns occur where contact with the hermaphrodite is never lost, sloppy turns involve a brief loss of contact that is regained, and missed turns are where lost contact is never regained. We recorded mating behaviour of male worms of various genotypes using a worm behavioural tracker (Yemini et al, 2013). In general, our data show that most turns are "good," and that "sloppy" or "missed" turns are relatively infrequent. *lury-1* and *npr-22* mutant animals did not show statistically significant differences in proportion of the different turning categories (Fig 4B); however, we did see that *lury-1* overexpression led to a significantly higher proportion of missed turns compared with control wild-type males (Fig 4C). This higher proportion of missed turns was not observed when *lury-1* overexpressors were crossed into *lury-1*–or *npr-22*–putative null mutants, indicating that knocking out the endogenous *lury-1* gene or that of the LURY-1 receptor, rescues this phenotype. These data indicate that, in this context, increased transcription of the *lury-1* gene modulates turning during male mating.

Exploratory (mate-searching) behaviour in *C. elegans* males is modulated by multiple internal and external cues including fed state, the presence of mates, and possession of an intact gonad (Lipton et al, 2004; Barrios et al, 2012; Wexler et al, 2020). Because we showed that LURY-1 peptides modulate both feeding and mating behaviours in male worms, we sought to better define how the presence of food affects mate-seeking in *lury-1*–mutant worms. First, we asked if male exploratory behaviour is impacted by lack of *lury-1*. Males leave food more quickly in the absence of mates than when mates are present, indicating that mate seeking takes priority over nutrition (Lipton et al, 2004). We performed a food leaving

assay using well-fed, non-mated control (*him-5(–)*), and *lury-1* mutant males, according to the protocol in Wexler et al (2020) (Fig 5Ai). By monitoring tracks made by male worms over a 24-h period, we found that *lury-1*–mutant males were more likely to move further away from food at earlier time points compared with control males, with statistically significant effects observed 6 h and onwards after the start of the assay (Figs 5Aii and iii and S3). This suggests that *lury-1* mutants place a greater priority on mate-seeking exploration versus feeding, compared with control males. We next tested whether the nutritional (well-fed or fasted) status of *lury-1*–mutant males affects mating efficiency (Fig 5B). Previous reports indicate that fasting male worms increases the likelihood that they will remain on the food patch, in contrast to well-fed males that rapidly leave food to search for mates (Lipton et al, 2004), suggesting that fed status modulates a behavioural transition between prioritising food-seeking versus mate-seeking. Using the *fog-2(–)* mating assay (Fig 3A), we found that fasting the male worms for 16 h before exposure to *fog-2(–)* hermaphrodites had no statistically significant impact on mating in control or *lury-1*–mutant males (Fig 5Bii). Taken together, these data indicate that although *lury-1*–mutant males overall displayed more food-leaving compared with controls, fed status did not have an overall impact on how mating efficiency is modulated by LURY-1.

Expression of *lury-1* was observed in neurons in the male tail that become active during the mating process (Susoy et al, 2021), as well as pharyngeal neurons (Figs 1A and B and S1). To determine whether these *lury-1*–expressing cells in the male worm are required for critical aspects of male mating, we performed genetic ablation of these cells by overexpressing caspase proteins (Chelur & Chalfie, 2007) using the *lury-1* promoter, and testing mating efficiency (Fig

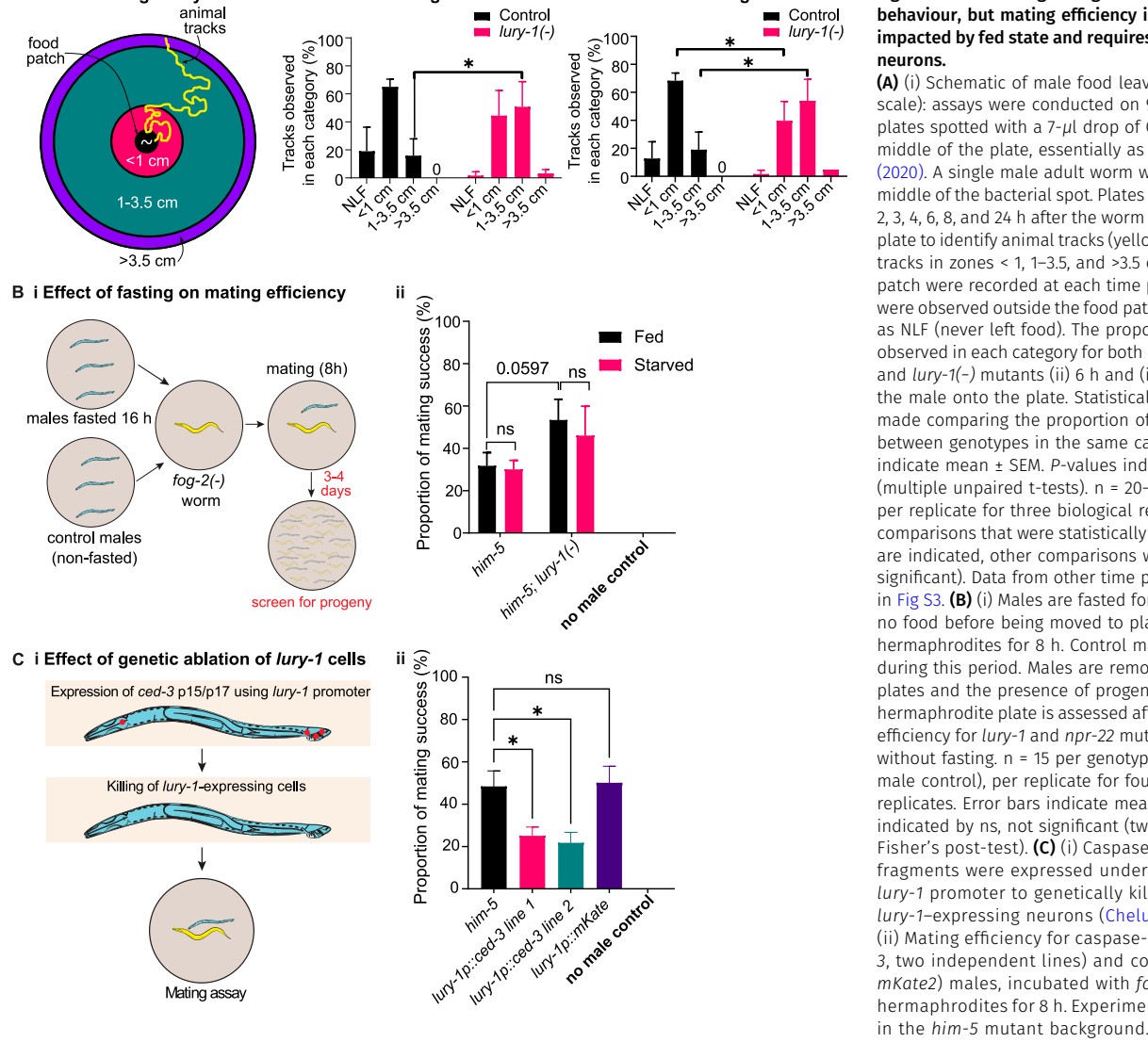

**Figure 5. LURY-1 signalling modulates food leaving behaviour, but mating efficiency is not strongly impacted by fed state and requires lury-1–expressing neurons.**
**(A)** (i) Schematic of male food leaving assay (not to scale): assays were conducted on 9-cm NGM agar plates spotted with a 7-µl drop of OP50 bacteria in the middle of the plate, essentially as in Wexler et al (2020). A single male adult worm was picked onto the middle of the bacterial spot. Plates were monitored at 1, 2, 3, 4, 6, 8, and 24 h after the worm was added to the plate to identify animal tracks (yellow). The presence of tracks in zones < 1, 1–3.5, and >3.5 cm outside the food patch were recorded at each time point. If no tracks were observed outside the food patch, this was marked as NLF (never left food). The proportion of tracks observed in each category for both control (*him-5*(−)) and *lury-1*(−) mutants (ii) 6 h and (iii) 8 h after placing the male onto the plate. Statistical comparisons were made comparing the proportion of tracks observed between genotypes in the same category. Error bars indicate mean ± SEM. *P*-values indicated by * < 0.05 (multiple unpaired t-tests). n = 20–21 per genotype per replicate for three biological replicates. Only comparisons that were statistically significant (*P* < 0.05) are indicated, other comparisons were ns (not significant). Data from other time points are indicated in Fig S3. **(B)** (i) Males are fasted for 16 h on plates with no food before being moved to plates with *fog-2*(−) hermaphrodites for 8 h. Control males are not fasted during this period. Males are removed from *fog-2*(−) plates and the presence of progeny on the *fog-2*(−) hermaphrodite plate is assessed after 3–4 d. (ii) Mating efficiency for *lury-1* and *npr-22* mutant strains with and without fasting. n = 15 per genotype (n = 10 for no male control), per replicate for four biological replicates. Error bars indicate mean ± SEM. *P*-values indicated by ns, not significant (two-way ANOVA, Fisher's post-test). **(C)** (i) Caspase *ced-3* p15 and p17 fragments were expressed under the control of the *lury-1* promoter to genetically kill *lury-1*–expressing neurons (Chelur & Chalfie, 2007). (ii) Mating efficiency for caspase-killed (*lury-1*p::*ced-3*, two independent lines) and control (*lury-1*p::*mKate2*) males, incubated with *fog-2*(−) hermaphrodites for 8 h. Experiments were performed in the *him-5* mutant background. n = 15 per genotype (n = 10 for no male control), per replicate for four biological replicates. Error bars indicate mean ± SEM. *P*-values indicated by ns = not significant. *P < 0.05 (one-way ANOVA, Fisher's post-test).

5Ci). We found that genetic ablation of *lury-1* cells led to significantly reduced mating efficiency compared with control *him-5* mutant males (Fig 5Cii). We also tested a control transgenic line in which *mKate2* was expressed using the *lury-1* promoter and found that this line showed a similar behaviour as control *him-5* males, demonstrating that expression of a non-active transgene using the *lury-1* promoter does not significantly affect male mating (Fig 5Cii). These findings indicate that the cells expressing *lury-1* in the male (pharyngeal neurons, male tail cells) are likely to function in promoting male reproductive behaviours.

## Discussion

In summary, we showed that the *lury-1* neuropeptide precursor is expressed in pharyngeal (feeding) neurons in both *C. elegans* sexes,

but with males showing further expression in tail cells required for mating. Interestingly, *lury-1*–mutant males show opposing phenotypes in pharyngeal pumping (feeding) behaviours compared with hermaphrodites: LURY-1 in males signals to increase pharyngeal pumping, which is the opposite to its effects in hermaphrodites (Ohno et al, 2017). In addition, LURY-1 influences male mating behaviour, with *lury-1*–mutant males showing higher mating efficiency than controls. This modulation of male mating requires *lury-1*–expressing neurons and is not affected by prior starvation, despite *lury-1* mutants showing increased food-leaving exploration behaviour. Based on these data, we propose that signalling via LURY-1 peptides may act to regulate both feeding and mating behaviours in a sexually dimorphic manner in *C. elegans*.

Luqin-type neuropeptides are evolutionarily ancient, with the origin of this peptide signalling system predating the divergence of protostomes and deuterostomes (Jekely, 2013; Mirabeau & Joly,

2013). Indeed, luqin peptides have also been implicated in feeding and reproductive behaviours in other invertebrates (reviewed in Yanez-Guerra and Elphick [2020]). Our study provides new information on the role of LURY-1 peptides in male *C. elegans*. There are, however, several questions that remain to be addressed: Firstly, the effects of *lury-1* overexpression are not as consistent as effects of the null mutation. Although *lury-1*–mutant hermaphrodites and *lury-1* overexpressing hermaphrodites show opposing phenotypes, *lury-1* mutant males and *lury-1* overexpressing males show effects in the same direction (both show less pharyngeal pumping) (Fig 2). In addition, *lury-1* overexpressing males did not show significant differences in mating efficiency compared with controls in the *fog-2(–)* assay, whereas *lury-1* mutants showed increased mating efficiency (Figs 3 and S3). One possible reason for this is that increasing transcription from the *lury-1* gene may impact the expression of other neuropeptides, with the profile of neuropeptides affected differing between different neuron subtypes (pharyngeal neurons vs male tail neurons). This could then result in different impacts of *lury-1* transgenic overexpression on different behaviours, that is, feeding and reproduction. Moreover, the increased transcription of the peptide precursor may cause excess LURY-1 to be present outside its physiological context, which may lead to other impacts on the biochemistry of these neurons. This is one possible explanation for why *lury-1* overexpression effects on male pharyngeal pumping appear to be independent of NPR-22 (Fig 2D). Alternatively, LURY-1 peptides may be acting through receptors other than NPR-22, which remain to be identified.

Secondly, specifically how LURY-1 release from the neurons of the male tail modulates feeding or mating has not yet been delineated. Recent research using whole-brain imaging and simultaneous behavioural tracking has begun to link individual male tail neurons to specific behaviours during mating (Susoy et al, 2021). Further characterising the functions of *lury-1*–expressing male tail neurons may reveal how LURY-1 release from these neurons, and its impacts on mating behaviour, are coordinated with release of the same signal from M1 and M2 pharyngeal neurons, not previously shown to affect male mating. We found that LURY-1 is expressed in male tail neurons including DX1, DX2, and DX3, which become active during the "turning" component of the male mating process, and that are also synaptically connected to male-specific sensory neurons (HOA, HOB, ray neurons, and PCA/PBC/PCC neurons) involved in other male mating behaviours. It is possible that *lury-1*–expressing cells are used in feedback loops with these sensory neurons during male mating, and that LURY-1 release from DX1, DX2, and DX3 cells in turn regulates the timing or effectiveness of other male mating behaviours. How release of LURY-1 from male tail neurons is coordinated with release from pharyngeal neurons, and how this impacts behaviour, is not yet clear. There are, however, several reports of pharynx activity being linked to male copulation: for example, pharyngeal pumping rate has been shown to decrease substantially in male *C. elegans* upon contact with the hermaphrodite vulva (Liu & Sternberg, 1995; Gruninger et al, 2006). In addition, the pharynx muscle and the pharyngeal neurosecretory neuron NSM were reported to influence male reproductive behaviours by acting on spicule protraction (Gruninger et al, 2006). These data, together with our findings, suggest that signals originating from pharyngeal muscle and

neurons influence male mating through a not yet fully delineated mechanism.

The sexually dimorphic expression pattern and function of LURY-1 could reflect differences in reproductive behaviours between the two sexes. Hermaphrodites can self-fertilise and therefore do not require mates to reproduce, whereas males only reproduce by mating. Moreover, male *C. elegans* will prioritise mate-seeking over remaining on a source of food, eventually abandoning food to search for mates—a behaviour that hermaphrodites do not exhibit (Lipton et al, 2004; Ryan et al, 2014). LURY-1 in hermaphrodites regulates feeding and egg-laying in a food-dependent manner (Ohno et al, 2017). Our data show that, in the absence of mates, lack of LURY-1 promotes male exploration away from food (Figs 5A and S3). This suggests that LURY-1 peptides normally signal the male to remain on food longer instead of leaving to search for mates. However, we also found that nutritional status (whether males are well-fed or fasted) does not strongly affect the modulation of mating efficiency by LURY-1 (Fig 5B). These two observations could occur either because (1) the mating efficiency assay as performed in this study is not as sensitive as the food-leaving assay, or (2) LURY-1 effects on mating efficiency result from a combination of food signals and other (internal) signals that remain to be defined. However, we cannot rule out a role for LURY-1 specifically in modulating locomotor activity (movement speed) independent of sensory signals from food or the presence of mates.

Male food-leaving as a strategy for seeking mates involves the coordination of external and internal signals including reproductive and nutritional states, as well as neuromodulatory signalling (Lipton et al, 2004; Barrios, 2014). Previous research has demonstrated that male food-leaving requires pigment-dispersing factor (PDF) peptides released from sex-shared (Barrios et al, 2012) and male-specific (Sammut et al, 2015) neurons. In addition, nematocin (NTC-1), an oxytocin/vasopressin-related neuropeptide in *C. elegans* (Beets et al, 2012; Garrison et al, 2012), modulates food leaving behaviour and coordination of the multi-step mating behaviour in males (Garrison et al, 2012). In addition to neuropeptides, male food-leaving is also impacted by serotonin and insulin signalling (Lipton et al, 2004), as well as dafachronic acid (DA) signalling through the nuclear hormone receptor DAF-12 (Kleemann et al, 2008). Lastly, the *lury-1*–expressing neurons identified in this study (M1, M2, DX1, DX2, and DX3) are all cholinergic (Pereira et al, 2015; Serrano-Saiz et al, 2017), suggesting that co-transmission of acetylcholine and LURY-1 peptides from these sex-shared and sex-specific neurons may permit modulation of male behaviours in multiple ways. How this co-transmission coordinates complex behaviours in the male worm is a potential topic for future study.

Here we have described another peptidergic signal, LURY-1, expressed both in neurons that regulate feeding and those that regulate mating, that may also modulate male copulatory behaviours. Future studies aiming to identify the combined behavioural impact of individual neuromodulator signals on neural circuits in the *C. elegans* male tail will provide a tractable model of how synaptic and "extrasynaptic" signals work together to drive sexually dimorphic behaviours. This may then help to elucidate the general principles through which sexually dimorphic behaviours arise from

the combination of neuronal signalling, neuroanatomy, hormonal influence, and experience, relevant to higher organisms.

# Materials and Methods

### Strain maintenance

Strains were maintained on NGM (nematode growth medium) plates seeded with *Escherichia coli* strain OP50 according to standard experimental procedures (Brenner, 1974). Day 1 adult *C. elegans* were used for all experiments. All experiments were performed at room temperature (22–23°C) and cultivated at 22°C.

Mutant strains used include Bristol N2 (wild-type), *fog-2(q71)*, *him-5(e1490)*, *lury-1(gk961835)*, *npr-22(ok1598)*, *him-5(e1490);lury-1(gk961835)*, and *him-5(e1490);npr-22(ok1598)*. All mutant strains were backcrossed >4 times to N2. A list of transgenic lines used, including full genotype information, is in Table S1. Transgenes were generated through standard microinjection procedures as in Chew et al (2018a, 2018b).

Strain LX2073 (*npr-22::GFP* fosmid) was a kind gift from the Koelle laboratory (Yale School of Medicine). Strains used for cell identification, NeuroPAL and P*lury-1::Venus*, were generously provided by Prof Oliver Hobert (Columbia University) and Prof Yuichi Iino (The University of Tokyo), respectively. Strains YLC124 (*him-5(e1490); pepEx011[Plury-1(3.4 kb)::ced-3 (p15)::nz::unc-54 3′UTR; Plury-1(3.4):: cz::ced-3 (p17)::unc-54 3′UTR; Punc-122::gfp::unc-54 3′UTR]*) and YLC125 (*him-5(e1490);pepEx012[Plury-1(3.4 kb)::ced-3 (p15)::nz::unc-54 3′UTR; Plury-1(3.4)::cz::ced-3 (p17)::unc-54 3′UTR; Punc-122::gfp:: unc-54 3′UTR]*) and YLC132 (*him-5;pepEx013[Plury-1(3.4 kb)::mKate2, unc-122::gfp]*) were generated by SUNY Biotech. Genotypes were confirmed via PCR.

Transgenes were cloned using the Multisite Gateway Three-Fragment cloning system (12537-023; Invitrogen) into pDESTR4R3 II, or using HiFi cloning (#E2621; NEB). Promoters for the *lury-1* gene were cloned either 2 or 3.4 kb before the ATG. For transgenes including *lury-1* genomic DNA spanning the coding sequence and 3′UTR, the entire coding region was cloned from the ATG start codon to the TGA stop codon, with an additional 330 bp in the UTR. Lines YL124 and YLC125 were generated by injection of plasmids containing the *ced-3* caspase fragments p15 and p17, expressed under the control of the *lury-1* promoter (3.4 kb). These constructs were generated by replacing the *mec-18* promoter in plasmids TU#806 (#16080; Addgene) and TU#807 (#16081; Addgene) (Chelur & Chalfie, 2007) with the *lury-1* promoter using Hifi cloning.

### Behavioural tests

#### Mating assay

This assay was performed according to the protocol outlined in Murray et al (2011) with the following modifications: *fog-2(q71)* hermaphrodites, which do not produce self-progeny, are picked at the fourth larval stage (L4) 16 h before the assay is conducted. Male *C. elegans* used in this assay contain a mutation in *him-5* that strongly potentiates the formation of male progeny to ~35% (compared with ~0.1% male progeny in wild-type populations). Male

worms are also picked at the fourth larval stage (L4) 16 h before the assay to an OP50-seeded plate with no hermaphrodites present. For the assay, one adult hermaphrodite was transferred onto an individual OP50 seeded plate and one adult male was picked to each plate to begin the mating assay. Males were removed from the plates after 3 or 8 h, as indicated. The plates were scored as successful or unsuccessful mating based on the presence or absence of progeny on the plates after >1 d. Sample size = n > 10 males per biological replicate for >3 replicates.

For mating assays combined with a period of starvation, assays were adapted as follows: L4 males were picked onto plates with food, or with no food, and left at 22°C for 16 h before being placed together with the *fog-2(q71)* hermaphrodite. Males were removed from the *fog-2* plates after 8 h.

#### Pharyngeal pumping

The rate of pharyngeal pumping was determined as previously described (Ohno et al, 2017). Adult animals were transferred onto fresh OP50-seeded plates and allowed to acclimatise for 30 min. After 30 min, the number of movements of the grinder in the posterior bulb of the pharynx observed in 1 min was counted (pumps/min). Male *C. elegans* used in this assay also contain a mutation in *him-5*. Sample size per genotype n > 10 per biological replicate for at least three replicates.

#### Turn scoring assay

Preparation of plates and recording procedures were performed as in Yemini et al (2013) and Yan et al (2017) with the following changes: video recordings were performed using a single *unc-13* hermaphrodite placed on food on a 3 cm NGM plate. *unc-13* hermaphrodites were used for ease of tracking—these animals have severely uncoordinated locomotion and do not move across the plate, allowing us to track only the freely moving (male) worm. After ~5 min, a male worm was picked onto this plate. Plates were placed on the tracker and recordings were started 15 min later. Each recording was 30 min. Genotypes were blinded during recording and analysis. Scoring the number of Good, Sloppy and Missed turns was performed according to Loer and Kenyon (1993), with the following conditions: male was in contact with hermaphrodite for >5 min for the duration of the recording, 1–3 contacts were recorded per animal. n > 10 per genotype.

#### Food leaving assay

Assays were conducted on 9 cm NGM agar plates spotted with a 7-µl drop of OP50 bacteria (OD600 = 1.0) in the middle of the plate, essentially as described in Wexler et al (2020). A single male adult worm was carefully picked onto the plate, in the middle of bacterial spot, using an eyelash pick. Before the assay, at least 20 L4 male worms were placed onto single-sex OP50-seeded plates for 16 h. During the assay, plates were monitored at 1, 2, 3, 4, 6, 8, and 24 h after the worm was added to the plate to identify animal tracks using a stereomicroscope with diascopic stage. The presence of tracks in zones <1, 1–3.5, and >3.5 cm outside the food patch were recorded at each time point. If no tracks were observed outside the food patch, this was marked as NLF (never left food). Care was taken not to excessively interact with the plates (to avoid tap vibrations) during the assay. 15 ml NGM plates in 9

cm petri dishes were poured 72 h before the assay, and were spotted with OP50 bacteria 16 h before the assay. 20–21 male worms per genotype were assayed per replicate for three biological replicates.

### In vitro GPCR activation assays

Cell-based activation assays were performed as described (Beets et al, 2012). NPR-22 cDNA was cloned into the pcDNA3.1 TOPO expression vector (Thermo Fisher Scientific). Receptor activation was studied in Chinese hamster ovary cells (CHO) stably expressing apo-aequorin (mtAEQ) targeted to the mitochondria as well as the human Gα16 subunit. The CHO-K1 cell line (ES-000-A24; PerkinElmer) was used for receptor activation assays. CHO/mtAEQ/Gα16 cells were transiently transfected with the NPR-22 cDNA construct or the empty pcDNA3.1 vector using the Lipofectamine transfection reagent (Thermo Fisher Scientific). Cells expressing the receptor were shifted to 28°C 1 d later, and collected 2 d post-transfection in BSA medium (DMEM/HAM's F12 with 15 mM Hepes, without phenol red, and 0.1% BSA) loaded with 5 $\mu$M coelenterazine h (Thermo Fisher Scientific) for 4 h to reconstitute the holo-enzyme aequorin. Cells (25,000 cells/well) were exposed to synthetic peptides in BSA medium, and aequorin bioluminescence was recorded for 30 s on a MicroBeta LumiJet luminometer (PerkinElmer) in quadruplicate. For dose–response evaluations, after 30 s of ligand-stimulated calcium measurements, Triton X-100 (0.1%) was added to the well to obtain a measure of the maximum cell $Ca^{2+}$ response. BSA medium without peptides was used as a negative control and 1 $\mu$M ATP was used to check the functional response of the cells. Cells transfected with the empty vector were used as a negative control (not shown). EC50 values were calculated from dose–response curves, constructed using a nonlinear regression analysis, with sigmoidal dose–response equation (Prism 9.0).

### Confocal microscopy

Worms for microscopy were picked onto 2% agarose pads (in milliQ water) and anesthetised with 75 mM sodium azide (in M9 buffer). At least 10 worms were imaged per genotype to obtain representative images. Imaging experiments using the NeuroPAL line were conducted using a Zeiss LSM880 as previously shown (Yemini et al, 2021), with settings available for download on yeminilab.com. Analysis of NeuroPAL images for cell identification was conducted as in Yemini et al (2021). Confocal images were obtained using a Leica SP8 confocal microscope (University of Wollongong) and Zeiss LSM880 (Flinders University). Z-stacks were analysed using Fiji (ImageJ).

### Statistical analyses

Statistical analysis for all experiments was performed using GraphPad Prism 8.0. In general, where two groups were compared, an unpaired *t* test was used (for the food leaving assay, multiple unpaired *t* tests were used). Where multiple groups tested with a single condition were compared, a one-way ANOVA with Fisher's post-test was used. Where multiple groups tested

with multiple conditions were compared, a two-way ANOVA with Fisher's post-test was used. The $\alpha$ value for all analyses is set at 0.05.

## Data Availability

The source data from this publication, including confocal micrographs used for expression analysis and cell identification, videos for male turning behavioural analysis, and raw data from behavioural studies, have been deposited to the Dryad database (https://doi.org/10.5061/dryad.cz8w9gj61). Reagents including *C. elegans* strains and expression plasmids are available from the corresponding author upon request.

## Supplementary Information

## Acknowledgements

Many thanks go to the members of the Chew and Schafer labs for helpful discussions in the preparation of this manuscript. We specifically thank Aelon Rahmani for proofreading the manuscript. We acknowledge the Flinders University Microscopy and Microanalysis facility for assistance with confocal microscopy and are particularly grateful to Dr Nicholas Eyre for assistance with NeuroPAL imaging. We gratefully acknowledge the *Caenorhabditis* Genetics Centre, which is supported by the National Institutes of Health (P40 OD010440), for providing some of the strains used in this study. We thank Prof Michael Koelle (Yale School of Medicine) for providing strain LX2073 (NPR-22 fosmid line), Prof Oliver Hobert for strain OH16230 (NeuroPAL), as well as Prof Yuichi Iino and Dr Hayao Ohno for strain JN2443 (P*lury-1*::Venus). I Beets is supported by the Research Foundation—Flanders (FWO G093419N). WR Schafer is funded by the Medical Research Council (MC-A023-5PB91), Wellcome Trust (WT103784MA), and the National Institutes of Health (R01 NS110391). YL Chew is funded by the National Health and Medical Research Council (NHMRC) (GNT1173448), the Rebecca L Cooper Medical Research Foundation (PG2020652), and the Flinders Foundation (Mary Overton Senior Research Fellowship).

### Author Contributions

MJ Gadenne: conceptualization, data curation, formal analysis, validation, investigation, methodology, and writing—original draft.
I Hardege: conceptualization, data curation, formal analysis, supervision, validation, investigation, methodology, and writing—original draft, review, and editing.
E Yemini: data curation, formal analysis, validation, and methodology.
D Suleski: data curation, formal analysis, and investigation.
P Jaggers: data curation, formal analysis, and investigation.
I Beets: data curation, formal analysis, funding acquisition, investigation, and methodology.
WR Schafer: conceptualization, formal analysis, supervision, funding acquisition, project administration, and writing—original draft, review, and editing.

YL Chew: conceptualization, data curation, formal analysis, supervision, funding acquisition, investigation, methodology, project administration, and writing—original draft, review, and editing.

## Conflict of Interest Statement

The authors declare that they have no conflict of interest.

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
