## [Reviewer comments · Life Science Alliance]

Life Science Alliance

Neuropeptide signalling shapes feeding and reproductive behaviours in male *C. elegans*

Matthew Gadenne, Iris Hardege, Eviatar Yemini, Djordji Suleski, Paris Jagggers, Isabel Beets, William Schafer, and Yee Chew
DOI: <https://doi.org/10.26508/lsa.202201420>

Corresponding author(s): Yee Chew, Flinders University

Review Timeline:

Submission Date:	2022-02-22
Editorial Decision:	2022-02-22
Revision Received:	2022-05-17
Editorial Decision:	2022-06-02
Revision Received:	2022-06-03
Accepted:	2022-06-03

Transaction Report:

Please note that the manuscript was reviewed at Review Commons and these reports were taken into account in the decision-making process at Life Science Alliance.

February 22, 2022

Re: Life Science Alliance manuscript #LSA-2022-01420

Dr. Yee Lian Chew
Flinders University
5D317, Flinders Medical Centre
Flinders Drive
Bedford Park, South Australia 5042

Dear Dr. Chew,

Thank you for submitting your manuscript entitled "Neuropeptide signalling shapes feeding and reproductive behaviours in male *C. elegans*" to Life Science Alliance. We invite you to re-submit the manuscript, revised according to your Revision Plan.

Thank you for this interesting contribution to Life Science Alliance. We are looking forward to receiving your revised manuscript.

Sincerely,

B. MANUSCRIPT ORGANIZATION AND FORMATTING:

Response to Reviewers Letter

LSA-2022-01420: *Neuropeptide signalling shapes feeding and reproductive behaviours in male C. elegans*

We are grateful to both reviewers for their considered and helpful comments. We have revised the manuscript according to these comments and have performed all suggested experiments, namely the food-leaving assay and cell identification of male tail *lury-1* neurons, as well as an additional control experiment including both *lury-1* mutant and overexpressor strains in the same assay. Our revised manuscript also contains additional discussion points as requested. All changes are indicated in yellow highlighting in the text. Please see below for a point-by-point response to all comments.

Reviewer comments:

Reviewer #1 (Evidence, reproducibility and clarity (Required)):

In this work, Gadenne et al. identify sexually dimorphic roles for the neuropeptide LURY-1 in the regulation of feeding and reproduction in *C. elegans*.

Through biochemical analysis of ligand and receptor pairs, reporter transgenes and behavioural analysis of overexpression and loss-of-function mutants, they show that LURY-1 normally acts through NPR-22 to inhibit pharyngeal pumping in hermaphrodites and to activate pharyngeal pumping in males while suppressing mating. Some of these results (i.e. identification of the receptor and the overexpression phenotype on hermaphrodites) recapitulate those previously published by Ohno et al. In addition, the authors extend the findings to the role that this neuropeptide system plays in the male.

Overall, most of the results are solid and the conclusions are supported by the experiments. There are still a few aspects that need addressing.

- One important message of the work is that *lury-1* regulates the trade-off between feeding and reproduction in males since the normal function of the gene (based on loss-of-function data) appears to increase pharyngeal pumping and reduce mating/intromissions. Regarding this, I think the authors should revisit some of the experiments exploring this trade off. They test whether starvation has any impact using the offspring production/mating assay and find no effect. However, this is not a very sensitive assay. Why don't the authors test the impact of starvation in the turning assay? Or better yet, and since they discuss it at length in the discussion, why don't they test the food-leaving assay in *lury-1* mutants?

This is an important suggestion. We have addressed this by performing **food-leaving experiments for control and *lury-1(-)* mutant males** according to the protocol of (Wexler...Portman, 2020, Current Biology). We found that well-fed *lury-1* mutant males move further away from food at earlier time points compared with controls (more exploration). Food leaving assay data has been added to **Figure 5A** and **Figure S3**.

We discuss these findings on **page 24-25** of the Discussion: "Our data shows that, in the absence of mates, lack of LURY-1 promotes male exploration away from food (**Figure 5A**, **Figure S3**). This suggests that LURY-1 peptides normally signal the male to remain on food longer instead of leaving to search for mates. However, we also found that nutritional status

(whether males are well-fed or fasted) does not strongly affect the modulation of mating efficiency by LURY-1 (**Figure 5B**). These two observations could occur either because 1) the mating efficiency assay as performed in this study is not as sensitive as the food-leaving assay, or 2) LURY-1 effects on mating efficiency result from a combination of food signals and other (internal) signals that remain to be defined. However, we cannot rule out a role for LURY-1 specifically in modulating locomotor activity (movement speed) independent of sensory signals from food or the presence of mates.”

In addition, they should integrate their findings (in the discussion) with those found by the Garcia lab showing an antagonistic interplay between pharyngeal pumping and spicule muscle excitability to regulate intromission.

Gruninger, T. R., Gualberto, D. G., LeBoeuf, B., & García, L. R. (2006). Integration of male mating and feeding behaviors in *Caenorhabditis elegans*. *The Journal of Neuroscience : the Official Journal of the Society for Neuroscience*, 26(1), 169-179.

This is an important point. We have integrated discussion of the findings from the Garcia lab in the context of our study to the Discussion (**page 24**).

“How release of LURY-1 from male tail neurons is coordinated with release from pharyngeal neurons, and how this impacts behaviour, is not yet clear. There are, however, several reports of pharynx activity being linked to male copulation: for example, pharyngeal pumping rate has been shown to decrease substantially in male *C. elegans* upon contact with the hermaphrodite vulva (Gruninger *et al*, 2006; Liu & Sternberg, 1995). In addition, the pharynx muscle and the pharyngeal neurosecretory neuron NSM were reported to influence male reproductive behaviours by acting on spicule protraction (Gruninger *et al.*, 2006)”

- In Fig 2. C, the fourth column is labelled as *lury-1 OE /lury- 1 -/-* . Is this correct or do the authors mean to show *lury-1 OE /npr-22 -/-* as in males in 2E?

Also, a statistical comparison between *lury-1 OE* and *lury-1 OE /npr-22 -/-* in males (2E) should be done because it seems like there is no significant difference, which would indicate that the LURY-1 OE effects are independent of the receptor NPR-22. This is particularly relevant in light of the observation that both OE and loss-of-function of LURY-1 result in the same inhibition of pharyngeal pumping phenotype in males.

We confirm that the labelling in **Fig 2C** is correct. We have added, as suggested, the statistical comparison between *lury-1 OE* and *lury-1 OE /npr-22 -/-* in males in **Fig 2D** (formerly Fig 2E), showing that the differences are not statistically significant. We have also added comments in the Discussion (**page 23**) acknowledging that some effects of the *lury-1 OE* appear to be independent of NPR-22.

To illustrate individual data points more clearly, we have changed the violin plots (indicating median and quartiles) for Figure 2 to scatter plots, with error bars indicating mean \pm SEM. This is also more suited to the ANOVA statistical test used, which compares means.

- The reporter transgene showing expression of LURY-1 uses a 3.4 kb promoter but the OE experiments (particularly pharyngeal pumping) are done with a 2kb promoter. The authors should show the expression pattern driven by the 2 kb promoter in order to better interpret the

results.

This is a good suggestion and we have included micrographs from the transgene expressing *lury-1* from a 2 kb promoter in the revised manuscript (**Figure S1**).

- Fig 3 - the mating efficiency of *him-5* is very variable, from less than 20 (in the set of experiments with mutant) to 50 % in the set of experiments with OE. This is in contrast to the mutant and OE worms which show a consistent value around 40%. The authors should run a few experiments comparing all three conditions (Control, mutant and OE) in parallel to better interpret the effects of OE and loss-of-function.

This is a good suggestion; we have added data from experiments comparing all three conditions (Control, mutant and OE) in parallel to **Figure S2Aiii** to better interpret the effects of *lury-1* over-expression and loss-of-function.

Reviewer #1 (Significance (Required)):

The finding that the *lury-1* neuropeptide system regulates the trade off between feeding and reproduction in a sexually dimorphic manner in *C. elegans* is interesting and important. The advance is limited however, since the reason/mechanisms for this difference are still to be elucidated

Reviewer #2 (Evidence, reproducibility and clarity (Required)):

Major comments:

none

Minor comments:

Overall, the paper is very well written and the data are very clear and the experiments properly controlled. I do not have any major criticism.

These minor comments should help the authors to make the presentation even clearer:

1) Is the identity of the peptide-expressing cells in the mail tail known? Can the connectome data be of any use in interpreting the sexually dimorphic effects of gene knockouts and overexpressions? The pharyngeal cells have been identified previously. There are also sex-specific full-animal connectomes and the mail tail connectome is fully described. Even though the signalling by the peptide may occur extrasynaptically, differences in synaptic connectivity may contribute to sexual dimorphism. The authors should attempt to interpret the results in light of the sex-specific synaptic connectomes. Alternatively, they should explain why it is not relevant or possible.

This is a great suggestion. To perform these analyses, we first identified *lury-1*-expressing neurons in the male tail using the tool NeuroPAL and cell ID maps published in Yemini et al., 2021, Cell and Tekieli et al., 2021, Development. As NeuroPAL can only be used together with green fluorescence reporter lines, we obtained the *lury-1p::Venus* line from Ohno et al., eLife 2017 to perform these analyses. Using *lury-1p::Venus*; NeuroPAL worms, we identified male-specific neurons **DX1, DX2 and DX3** as *lury-1*-expressing neurons in the male tail.

We used connectomics data for the male worm (Cook... Emmons, Nature, 2019) to identify the cells synaptically connected to these *lury-1*-expressing male tail neurons. DX1, DX2 and DX3 are male-specific neurons connected to multiple cells in the male tail, including hook sensillum neurons HOA/HOB, multiple ray neurons, and neurons that innervate the post-cloacal sensillum, all of which are male-specific and shown to be required for male mating behaviours. Taken together, these expression and connectivity data indicate that *lury-1*-expressing neurons in the male tail are likely to be involved in coordinating male-specific mating behaviour. We have incorporated these new data into our revised manuscript (**Figure 1B, Results page 5-6, Discussion page 23-24**).

2) Fig 1 legend: 'in pharyngeal neurons M1 and M2 (yellow arrows)' - I could not find the yellow arrows

The legend in the original manuscript referred to an older version of the figure, which has since been changed and the arrows removed.

3) 133 'a putative null mutation which encodes a 2.5 kb deletion in the npr-22 genomic locus

(deleting four of six exons)' - encodes a deletion sounds a bit strange to me - maybe simply 'null mutation with a 2.5 kb...'

We agree that the suggested phrasing sounds better and this sentence has been fixed (**Page 9**).

4) In Fig 2 panels B and C seem to show the same data for the N2 and lury-1(-) genotypes. Is there any difference between these two data sets? If not, panels B and C could be merged or lury-1(-) should not be shown twice (i.e. not on panel C).

The data for N2 and lury-1(-) are indeed the same for B and C, so we have merged the panels as suggested (**new Fig 2B**).

As mentioned above in our response to Reviewer 1, to illustrate individual data points more clearly, we have changed the violin plots (indicating median and quartiles) for Figure 2 to scatter plots, with error bars indicating mean \pm SEM. This is also more suited to the ANOVA statistical test used, which compares means.

5) In Fig 2 panel E the npr-22 genotype should be shown as npr-22(i) for consistency.

This has been fixed in **Fig 2D** (Formerly Fig 2E)

6) Fig 3 legend: 'n>10 per genotype' add that n here refers to the number of males

Thanks for pointing this out - this has been fixed in the **Fig 3 legend**

Reviewer #2 (Significance (Required)):

This is an interesting paper exploring the function of a luqin neuropeptide and its receptor in males and hermaphrodites of the nematode *C. elegans*. The authors carry out a careful genetic analysis and uncover sexually dimorphic roles for this neuropeptide signalling system. The results will be of interest to those working on neuropeptide signalling, sexual dimorphism and nematode feeding and mating behaviour.

I am an expert in neuropeptide signalling and invertebrate neural circuits.

June 2, 2022

RE: Life Science Alliance Manuscript #LSA-2022-01420R

Dr. Yee Lian Chew
Flinders University
5D317, Flinders Medical Centre
Flinders Drive
Bedford Park, South Australia 5042
Australia

Dear Dr. Chew,

Thank you for submitting your revised manuscript entitled "Neuropeptide signalling shapes feeding and reproductive behaviours in male *C. elegans*". We would be happy to publish your paper in Life Science Alliance pending final revisions necessary to meet our formatting guidelines.

- please check your figure legend for Figure 2; you have a panel E in the legend, but this is not in the figure
- please remove the panel A in Figure S2 because it is the only panel in the figure
- please adjust any figure callouts with the designation EV and update these to Supplementary Figures; for example, Figure EV3 should be Figure S3

A. FINAL FILES:

B. MANUSCRIPT ORGANIZATION AND FORMATTING:

Sincerely,

Reviewer #1 (Comments to the Authors (Required)):

The authors have addressed all my comments and the manuscript is much improved.
Regarding the identification of the DXs as the lury-1 -expressing neurons in the male tail, the authors may want to mention that these neurons have been previously suggested to be involved in sperm transfer (Liu, K. (1995). Sensory regulation of *C. elegans* male mating behaviour. Volume PhD in Biology (Pasadena, CA: California Institute of Technology), a role which may explain why lury-1 has an effect on mating success

Reviewer #2 (Comments to the Authors (Required)):

The authors have addressed all my comments.

June 3, 2022

RE: Life Science Alliance Manuscript #LSA-2022-01420RR

Dr. Yee Lian Chew
Flinders University
5D317, Flinders Medical Centre
Flinders Drive
Bedford Park, South Australia 5042
Australia

Dear Dr. Chew,

Thank you for submitting your Research Article entitled "Neuropeptide signalling shapes feeding and reproductive behaviours in male *C. elegans*". It is a pleasure to let you know that your manuscript is now accepted for publication in Life Science Alliance. Congratulations on this interesting work.

DISTRIBUTION OF MATERIALS:

Again, congratulations on a very nice paper. I hope you found the review process to be constructive and are pleased with how the manuscript was handled editorially. We look forward to future exciting submissions from your lab.

Sincerely,
